# Lightweight Internet of Things Botnet Detection Using One-Class Classification

**DOI:** 10.3390/s22103646

**Published:** 2022-05-10

**Authors:** Kainat Malik, Faisal Rehman, Tahir Maqsood, Saad Mustafa, Osman Khalid, Adnan Akhunzada

**Affiliations:** 1Department of Computer Science, COMSATS University Islamabad, Abbottabad 22060, Pakistan; kainatmalik055@gmail.com (K.M.); frehman@cuiatd.edu.pk (F.R.); tmaqsood@cuiatd.edu.pk (T.M.); saadmustafa@cuiatd.edu.pk (S.M.); osman@cuiatd.edu.pk (O.K.); 2Faculty of Computing and Informatics, University Malaysia Sabah, Kota Kinabalu 88400, Malaysia

**Keywords:** internet of things (IoT), one-class KNN, botnet detection, classification

## Abstract

Like smart phones, the recent years have seen an increased usage of internet of things (IoT) technology. IoT devices, being resource constrained due to smaller size, are vulnerable to various security threats. Recently, many distributed denial of service (DDoS) attacks generated with the help of IoT botnets affected the services of many websites. The destructive botnets need to be detected at the early stage of infection. Machine-learning models can be utilized for early detection of botnets. This paper proposes one-class classifier-based machine-learning solution for the detection of IoT botnets in a heterogeneous environment. The proposed one-class classifier, which is based on one-class KNN, can detect the IoT botnets at the early stage with high accuracy. The proposed machine-learning-based model is a lightweight solution that works by selecting the best features leveraging well-known filter and wrapper methods for feature selection. The proposed strategy is evaluated over different datasets collected from varying network scenarios. The experimental results reveal that the proposed technique shows improved performance, consistent across three different datasets used for evaluation.

## 1. Introduction

Internet of things (IoT) is an evolving technology that offers a platform for different gadgets to automate activities in various areas. The idea of the “Internet of Things” (IoT) is utilized in numerous fields of data and communication designing. According to a recent study presented in Ref. [1], the market size of worldwide IoT can reach USD 1567 billion by 2025. The ubiquitous nature and ease of use enables IoT devices to sense and transmit data for numerous advanced applications. It is anticipated that the number of IoT gadgets will reach 75 billion by 2030 [2].

IoT devices transmit data via communication mediums, which makes them vulnerable to various security attacks [3]. Traditional network security models designed for conventional networks become inefficient for IoT-based systems due to their low computational and storage capacities and inability to install frequent security updates.

Botnet is a threat that takes advantage of the security lapses of IoT. More precisely, botnets are networks consisting of nodes that were infected by malware, which turned them into bots that can attack any system as a response to orders of a botmaster. Many destructive DDoS attacks have been launched using IoT botnets; Mirai is one notable example. Mirai compromised 100,000 IoT devices in the span of just a few months and created a large botnet out of them. Mirai botnet was then used to generate the highest rated DDoS attacks of almost 1.2 Tbps causing Twitter and GitHub to go offline for their users [1].

With increasing number of threats and inability of IoT devices to enhance security, users cannot depend just on updates for each security vulnerability. Therefore, more consideration should be given to investigate security vulnerabilities, their identification, classification and recovery of IoT devices after the attacks. This makes an intrusion detection system (IDS) essential for IoT.

Network traffic generated from IoT devices follows specific patterns of variable nature depending upon the application and scenario. The traffic data can be given input to various machine-learning (ML) algorithms to perform traffic prediction [4]. Moreover, ML can be used as one of the most efficient computational models to be effectively used in the IoT gadgets for threat prediction and detection without human supervision [5,6].

The viability of machine-learning methods has empowered research scientists to utilize learning models in IoT domains to improve detection and identification among IoT gadgets. In Ref. [7], two machine-learning models were used to guard IoT sites against DDoS attacks. The proposed technique can identify as well as classify the attacks. For reaching a productive solution utilizing a machine-learning algorithm, an appropriate real-world dataset is exceptionally fundamental along with a feature selection method [8,9]. Although, a lot of work has been performed, there is still a need for an efficient feature selection technique based on an algorithm that can be trained on consumer IoT devices to effectively recognize IoT botnet attacks with an increased detection rate and reduced false alarms.

This paper proposes a model that offers a lightweight solution to detect anomalous behavior in resource-constrained IoT devices. The proposed solution makes use of the one-class K-nearest neighbor (KNN), which does not need anomalous data to train a model. One-class KNN is a specific machine-learning strategy in which, rather than grouping an instance in one of the different pre-characterized patterns, the instance is modeled as a single pattern and used to perceive whether a new sample belongs to that pattern or not. For reduction in computation power and ML decision time, feature selection (consisting of wrapper and filter method) is used along with one-class KNN, which has shown the best performance among one-class classifiers [10]. For gaining deep insight into features, a customized script of feature extraction is used. Feature space of datasets was reduced by almost 72%; only important features that had an impact on performance were chosen. This technique shows a satisfactory performance by achieving an F1-score of 98% to 99% on different IoT datasets.

The rest of the paper is organized as follows. Section 2 presents the literature review; Section 3 presents the methodology; and Section 4 discusses the experiment with one-class KNN. In Section 5, the results and analysis are presented. Lastly, Section 6 discusses the conclusions and future work.

## 2. Related Work

ML approaches are known to be efficient in securing IoT networks when used with appropriate datasets [11,12]. Many researchers used different supervised and unsupervised approaches for detecting and blocking attacks in IoT networks. In utilizing ML algorithms for the security of IoT networks, researchers have used synthetically generated datasets, e.g., CIDDS-001, UNSWNB15 and NSL-KDD [13], to name a few. A significant problem is finding appropriate datasets, as well as a proper feature selection method [1].

Authors in Ref. [14] used a combination of machine-learning strategies and parallel data processing for detecting IoT botnet attacks. For reducing the size of training data, preprocessing was applied to clean the raw data. Because of high computational cost, the proposed work could only be applied on high-energy nodes. Researchers in Ref. [15] developed a solution for the detection of IoT botnet using deep learning. Autoencoder was used to detect the malicious communication, which is computationally expensive. The suggested solution detected the IoT botnet with zero false positive rate (FPR). A total of 115 features were used to train the algorithms, causing an overhead on resources. In Ref. [16], the researchers suggested a framework based on ML to detect the IoT botnet in large networks with an accuracy of 94%. The multiclass algorithm was trained with the selected features. However, the specific details of the features were not provided. In Ref. [17], the authors suggested an IoT botnet detection framework based on host features. A machine-learning algorithm was trained using the synthetic dataset, making the solution less reliable. Moreover, feature selection was also not used; rather, the machine-learning algorithm was trained with numerous features. The dataset diversity was not taken into account, which is important to check the scalability of the system.

The authors in Ref. [18] used a supervised learning algorithm to secure a smart home by detecting intrusions. The proposed framework considered three cases. First, the IoT devices were grouped based on their conduct in ordinary scenarios. Second, during an attack, the malicious packets were distinguished based on their characteristics. Finally, the attack was classified based on its nature. In Ref. [19], the authors proposed a technique to secure a smart home by using a supervised algorithm (support vector machine) to distinguish between normal and abnormal behavior. In Ref. [20], the security of an IoT domain was provided by constantly monitoring network traffic. A total of 28 different gadgets were used to attain normal traffic behavior, and then, a classification algorithm was used to classify normal and abnormal traffic attributes.

In Ref. [21], the detection of different attacks, e.g., DDoS, data type probing, etc. was performed with the help of machine-learning algorithms. The performance of various algorithms was also assessed, and random forest was found to be effective. In Ref. [22], the proposed scheme used the dataset NSL-KDD, which is a generic dataset, not related to the IoT environments. Moreover, an ineffective feature selection method was used to reduce the redundancy of data. The authors in Ref. [7] designed a framework called flow guard, which can be used for the defense against IoT botnet DDoS attacks. Variation in traffic volume was used to detect a DDoS attack generated by the IoT botnet. Long short-term memory algorithm was used for the detection. The classification accuracy was about 99%.

Researchers in Ref. [23] proposed a deep-learning-based botnet detection in IoT for highly imbalanced datasets. For the improvement of the imbalanced class, additional samples of that class were generated. A real-world IoT dataset was used to train a deep recurrent neural network [24]. The proposed solution was able to detect the IoT botnet with a 99% of F1-score. The complexity due to deep-learning methods creates a deployment problem in IoT networks, as the devices are resource constrained. In Ref. [25], the authors proposed a decision-tree-based machine-learning approach to detect IoT botnets. A dimensionality reduction method was introduced to enhance the machine-learning performance. Results indicated that the decision tree combined with random projection was able to detect the IoT botnet with 100% accuracy. However, the dimensionality reduction process results in some amount of data loss. To optimize the detection accuracy of IoT botnets, ensemble learning was introduced by the researchers in Ref. [26]. The algorithms from supervised, unsupervised and regression learning were chosen to enhance accuracy and reduce the number of features at the time of training. Ensemble learning requires high computation power, as multiple models need to be built.

Most of the above-discussed techniques provided good solutions with an accuracy of more than 90%. However, they are either computationally expensive due to employing costly machine-learning models, or the selected dataset is generic and not specific to IoT environments. Moreover, selecting a large number of features affects the detection efficiency of an algorithm in IoT environments. Therefore, in the proposed technique, we present an efficient feature selection mechanism that renders it a lightweight solution for IoT devices. Moreover, to reduce the computational overhead, we utilized a one-class classifier to obtain a satisfactory detection rate along with low false-alarm rates.

## 3. Proposed Methodology

In the proposed work, we considered datasets generated in three different labs as mentioned in Refs [27,28,29]. A real consumer IoT gadget network was utilized to produce traffic in these labs, where real-time data of normal and malware traffic were captured. Three types of IoT botnets, Mirai, Bash lite and Torii, were deployed in the network. These three datasets are made publicly available by the researchers and are provided in PCAP formats. The PCAP formats are typically related to network analyzer programs, such as Wireshark. They generally contain the information extracted from packets within a network. The PCAP files can be analyzed to discover the network running state and can be used to detect any problem that occurred in the network. Moreover, the PCAP files can be utilized to search for the information transmitted between the hosts inside or outside of the network. The system architecture is shown in Figure 1, which represents the overall model of the proposed solution. The mechanism of the proposed solution is outlined in Algorithm 1. The methodology of the proposed work is divided into the following stages.
**Algorithm 1:** Botnet detection using one-class KNN**Input: datasets *d*_1_, *d*_2_, *d*_3_**1.Convert datasets *d*_1_, *d*_2_, *d*_3_ into PCAP format2.Apply filtering based on source and destination IP3.Perform features extraction for f∈f1,f2,f3,…fn4.Data preprocessing to eliminate missing, infinite, NAN and HEX values5.Perform feature selection using6.  Filter method7.  Wrapper method8.End feature selection9.For each dataset *d*_1_, *d*_2_, *d*_3_ apply one-class KNN10.  Load the training dataset11.  Choose the value of *k*12.  Train the model13.  Load the test dataset14.  For each point in the test data untilpoint *= NULL*15.    Find Euclidian distance *d* to all training data points d=∑i=1kxi−yi2
16.   Store *d* in a list *L* and sort it17.   Choose the first *k* points18.   Assign class to the test points19.  End For20. End For

### 3.1. Data Assortment

A real-time IoT device network traffic dataset was obtained from three different IoT labs [27,28,29] that contained benign and malicious traffic. Consumer IoT devices were used by the labs to generate the dataset. All the datasets were in PCAP file format, which gave us the ability to thoroughly analyze the data at the packet and frame level. The three datasets were labeled by their source, which helped in distinguishing the generated results for each dataset.

### 3.2. PCAP Filtering

PCAP filtering is a method to filter out specific packets. If the source IP is of interest, the source IP filter can be applied, and the packets will be shown with that source IP. PCAP files of each IoT device were filtered based on some specific filters. For this purpose, the broadly utilized network monitoring tool Wireshark was used. Wireshark is a free and open-source tool used to analyze network traffic. It captures the traffic progressively and stores it in a PCAP file for later examination. PCAP files are the files in which all IoT gadgets’ captured traffic is available. Once the PCAP file is loaded in Wireshark, it is reviewed for protocols and fields. The IP addresses of IoT gadgets are available in each dataset, so it was convenient to separate PCAP files for each IoT gadget. To separate traffic for an IoT gadget, a filter based on source and destination IP was applied. Steps for filtering PCAP files are shown in Figure 2.

### 3.3. Feature Extraction

The most basic and significant phase of this work was the extraction of features from PCAP files, which were filtered and labeled according to the gadget. These input features have a significant role in output variable prediction.

### 3.4. Shark Script

A bash script, named a Shark script, was programed for the purpose of feature extraction from PCAP files so that the network transmission characteristics and properties could be studied. The developed script was able to extract 46 features from a PCAP file using Tshark [30], which is a command line [31] version of Wireshark that can capture live traffic or investigate packets from PCAP files. It writes its output in a file using various kinds of filters that can be applied to assess a particular packet.

When a PCAP file was given as input to the Shark script, it produced a CSV file containing 46 predictor variables as shown in Table 1. The feature extraction process is shown in Figure 3.

### 3.5. Data Preprocessing

Data preprocessing is applied for cleaning and making the data suitable for the training of the ML classifier. Frequently, data contain some missing, invalid and infinite variables. These values should be removed or treated to make data available for an algorithm that can be trained on the data. The dataset available in CSV is passed through the data preprocessing phase as shown in Figure 4.

The dataset of each gadget present in the CSV file contains some missing values. The missing variable can be managed through various techniques. We substituted the missing values with a numeric zero by utilizing the Pandas library’s function [32].

#### 3.5.1. NaN and Infinite Value Checking

All the datasets were checked for the NaN and infinite values, as these contribute negatively to ML training. The Pandas library was utilized to address this issue.

#### 3.5.2. Categorical Values

Those variables, which have non-numeric values and are divided into groups, are called categorical variables. Most of the ML classifiers require input variables to be in numeric or number format. For this reason, categorical variables are converted into numerical values by utilizing encoding. In this paper, the “one-hot encoding” was utilized to convert all categorical values into numeric ones [33]. The “Get_dummies method” of Pandas was utilized for this purpose [34]. When the CSV dataset was inspected for categorical values, the source and destination IPs addresses were considered as categorical. These two were converted into numerical values utilizing the aforementioned library.

#### 3.5.3. Hex Values

The dataset present in CSV files contains a property ip.flags that is in hex format. All the features other than ip.flags are in the decimal number system. So, to keep all the features in the same number system, ip.flags feature was converted using Pandas library from hex to decimal number system so that it could be easily recognized by the classifier.

### 3.6. Feature Selection

After implementing feature extraction with the help of Shark script, 46 features were extracted from each dataset. Then, the feature selection process was applied to reduce the number of independent variables, thereby reducing the computational overhead. Feature selection process allows keeping only those variables that greatly affect the response variable and eliminates irrelevant or repetitive features. Training ML classifiers with a high number of independent variables not only increases the training time and memory utilization but also affects the performance of the algorithm. In our work, we applied the filter and wrapper methods of feature selection, as discussed below.

#### 3.6.1. Filter Method

Filter method uses statistical thresholds by which the features are dropped or kept. Filter method can be further split up into two primary classes: univariate filter method and multivariate filter method, as discussed in the subsequent text.

##### Univariate Filter

In the univariate filter strategy, single features are kept or dropped by measurable test by utilizing a threshold. The univariate filter strategy began with maximum features and, by applying diverse thresholds and filters, the number of features was dropped down. The univariate filter method was implemented by using the following steps:First, Jupyter Notebook was set up, and essential libraries were imported.Datasets that contained benign and malicious files were imported.Each feature was examined for missing value so that there should be no feature with a missing value in the dataset.Constant features were examined by setting the variance threshold equivalent to zero. A total of 14 features were found with constant values, and those features were discarded.Quasi-constant features were examined by adjusting the variance threshold to 0.1. A total of five features not meeting the criteria were deleted.Dataset was examined for repeated columns, and duplicate columns were deleted to make them distinct.At the end of the univariate filter method, the remaining features number 27.

##### Multivariate Filter

The multivariate filter method looks for a relationship among variables and eliminates repetitive features.

After implementing the univariate filter method, the remaining 27 features were inspected for correlation utilizing Pearson’s correlation coefficient using library phik_matrix [35].For visualizing the correlation score, a heatmap was plotted as shown in Figure 5.Features with a score correlation value greater than 0.95 were filtered. As a result, 5 more features were dropped, leaving the feature count at 22.Resulting data frame comprised 22 features and was saved as CSV for additional utilization and filtering through the wrapper method.

#### 3.6.2. Wrapper Method

In the wrapper method, the feature selection cycle depended on a particular ML algorithm that we attempted to fit on a given dataset. Moreover, in the wrapper method, we attempted to utilize a subset of features and train a model over those. Considering the derivations that we drew from the previous model, we applied the wrapper method to further add or eliminate features from our subset. The forward and backward phases of the wrapper method are discussed below.

##### Forward Feature Selection

This technique begins with no element and then adds a single feature in each turn.

##### Backward Feature Elimination

This technique begins with all features and eliminates one feature in each turn.

In the first step, we imported machine-learning libraries, loaded the dataset and split it into training and testing sets.We performed the forward feature selection method in Python. Random forest classifier was used for feature selection.All 22 features were input to random forest that assigned a score to each feature using the mean decrease accuracy measure.Among the features, 13 features were selected based on their importance. These features produced the highest accuracy with random forest.The selected features were then used to train our algorithm as shown in Table 2.We then utilized the selected features to construct a full model utilizing our training and test datasets and then evaluated the accuracies.

## 4. Experiments with One-Class KNN

The traffic of IoT devices is usually very specific and does not experience frequent fluctuations. For instance, a smart fan’s only task is to sense the temperature or some other on–off operation. This is why the benign network traffic behavior of IoT devices is predictable and does not change over time. The one-class KNN algorithm was chosen, as it generates a classification model that can be easily explained, allowing a better interpretation of the classification result and making the model a lightweight solution [36].

One-class KNN is a classification algorithm that takes a single class for training, and based on that training, it predicts whether the future dataset is normal or malicious. The single-class KNN classifier has various parameters that might be changed. For example, the quantity of neighbors can be changed. Therefore, the distances of normal *k* to the first *k* neighbors are determined. The threshold value for obtaining outlier classes can be changed. Likewise, the distance metric can be changed. The one-class KNN is formulated as follows. The distance between two samples, *x* and *y*, can be found by the following distance functions.
(1)$$\sqrt{\overset{k}{\underset{i=1}{\sum }}{\left(x_{i}-y_{i}\right)}^{2}\;\;\;\;\;}$$
(2)$$\overset{k}{\underset{i=1}{\sum }}\left|x_{i}-y_{i}\right|$$
(3)$${\left[\overset{k}{\underset{i=1}{\sum }}{\left(\left|x_{i}-y_{i}\right|\right)}^{q}\right]}^{\frac{1}{q}}$$

In the proposed research work, 80,000 (80K) instances of benign and malicious datasets were used, out of which 60K were benign and 20K were malicious. The benign instances were split by 80–20% ratio to train and test the algorithm performance. Because of recognizing whether a packet is malicious or normal, the classification is evaluated comparatively with the training dataset, generating four outputs:

### 4.1. True Positive

Packets are benign and are predicted benign.

### 4.2. False Positive

Packets are malicious but predicted benign.

### 4.3. True Negative

Packets are malicious and are predicted malicious.

### 4.4. False Negative

Packets are benign and are predicted malicious.

To test the classification performance, we utilized commonly used measures, which are accuracy, F-measure, precision and recall.

### 4.5. Precision

Precision (P) measures the total amount of anomalous packet recognition that was correct.
(4)$$\mathrm{Precision}=\frac{\mathrm{tp}}{\mathrm{tp}+\mathrm{fp}}\;$$

### 4.6. Recall

Recall estimates what number of anomalous packets were recognized correctly.
(5)$$\mathrm{Recall}=\frac{\mathrm{tp}}{\mathrm{tp}+\mathrm{fn}}$$

### 4.7. F-Measure

The F-measure is a proportion of a test’s accuracy. It is determined from the precision and recall of the test.
(6)$$F-\mathrm{measure}=\frac{2\times \mathrm{Precision}\times \mathrm{Recall}}{\mathrm{Precision}+\mathrm{Recall}}$$

The performance of one-class KNN classifier is analyzed based on F1-score. F1-score is appropriate for an imbalanced dataset and due to skewness property in class dispersing in the dataset [37], in which one class is dominating, and in our situation, the benign class was dominating.

## 5. Results and Analysis

Two different feature selection methods were used, filter and wrapper methods, which carry out operations to eliminate features that do not have any impact on performance in terms of reducing the classification time. Machine-learning models can be investigated utilizing more than one performance metric. As our dataset was imbalanced and asymmetric, it was suitable, in this situation, to utilize the F1-score as a performance metric.

### 5.1. Dataset Description

Three types of datasets were used that were collected from three different IoT labs as mentioned previously. The description of IoT devices is shown below in Table 3. For the MedBiot [24] dataset collection, 83 IoT devices were connected in the lab environment. Data were collected for both benign and malicious traffic. For malicious traffic, IoT botnets, such as Mirai, BashLite and Torii, were deployed. The dataset [25] consists of a labeled dataset, which included botnet activity and DDoS attacks.

A lab environment consisting of an IoT smart camera was set up. Benign and malicious traffic for the IoT camera was captured. HCRL lab [26] generated an academic purpose dataset, where benign dataset was captured for smart home devices, such as smart speakers. A malicious dataset was also collected after infecting the devices with IoT botnet binaries, such as Mirai.

### 5.2. Accuracy, Precision and Recall

Accuracy, precision and recall of different datasets are shown in Figure 6, Figure 7 and Figure 8 when these are trained and tested using one-class KNN. It can be seen that the proposed model outperforms the counterparts in terms of accuracy, precision, recall and F-measure by reducing the feature space. Additionally, the outcomes show that the proposed model is adaptable and exhibits consistent behavior in various IoT environments, as it was tested on three different datasets from different IoT domains. Previous researchers disregarded the significance of feature selection methods, which can decrease the number of features for the training phase. Therefore, in our case, the training with a smaller set of features decreases the training time and utilizes less computation power.

### 5.3. F1-Score on Different Datasets

F1-score against different datasets is shown in Figure 9. From the results, it can be analyzed that the proposed solution provides a consistent F1-score across three different IoT datasets. Thus, the proposed solution is able to detect the malicious pattern without the dependency on the environment.

### 5.4. Effect of Feature Selection

Feature selection methods are applied to reduce the features and shortlist those features that affect the performance of the machine-learning model. The following Table 4 shows the effect of feature selection on the F1-score.

Those features that have a lesser effect on performance are removed. After the filtering method, the wrapper method is applied, which selects top features for machine-learning training purposes, and this also improves the F1-score.

### 5.5. Comparison with Recent Research Works

The proposed solution is compared with the research [17,18], and results are shown in Table 5, which used different ML algorithms for evaluating botnet detection.

As shown in Table 5, the proposed solution has a good performance, attaining a mean F1-score of 99%, also shown in Figure 10, Figure 11, Figure 12 and Figure 13. The methods of feature selection (filter and wrapper) reduced the feature space and improved the performance of classifiers, which were not used in the existing research work. Moreover, feature extraction was performed using a shark Script that was able to extract 46 features out a PCAP file, which represents all the behavior of communication.

The combination of the Shark script dataset and the one-class KNN method gives the highest F1-score. The feature selection methods of filter and wrapper decreased the feature space by up to approx. 72%.

### 5.6. Comparison of Training Time before and after Feature Selection

It is shown in Table 6 that training and prediction time after applying feature selection was reduced, which shows the proposed model is a lightweight strategy.

## 6. Conclusions

In this paper, one-class KNN was deployed to detect deleterious IoT botnet in a heterogeneous environment. To overcome the computation power limitation, the lightweight solution was derived. Feature selection methods, such as filter and wrapper methods, were used to select the best features that had an impact on machine-learning classifier performance. Using the filter and wrapper methods, feature space was reduced by up to approx. 72% for different datasets. The suggested solution was environment independent, as it was tested in three different IoT environments that were using different IoT devices. The proposed solution was able to detect IoT botnet with an F1-score of 98% to 99% for different IoT botnet datasets. The solution was able to detect the unknown malicious traffic of IoT in different IoT environments.

As future work, we would like to detect other network malicious behaviors of IoT devices, such as man in the middle (MITM) and replay attacks. We would like to simulate a smart home environment to collect the dataset for benign as well as malicious traffic.

## Figures and Tables

**Figure 1 sensors-22-03646-f001:**
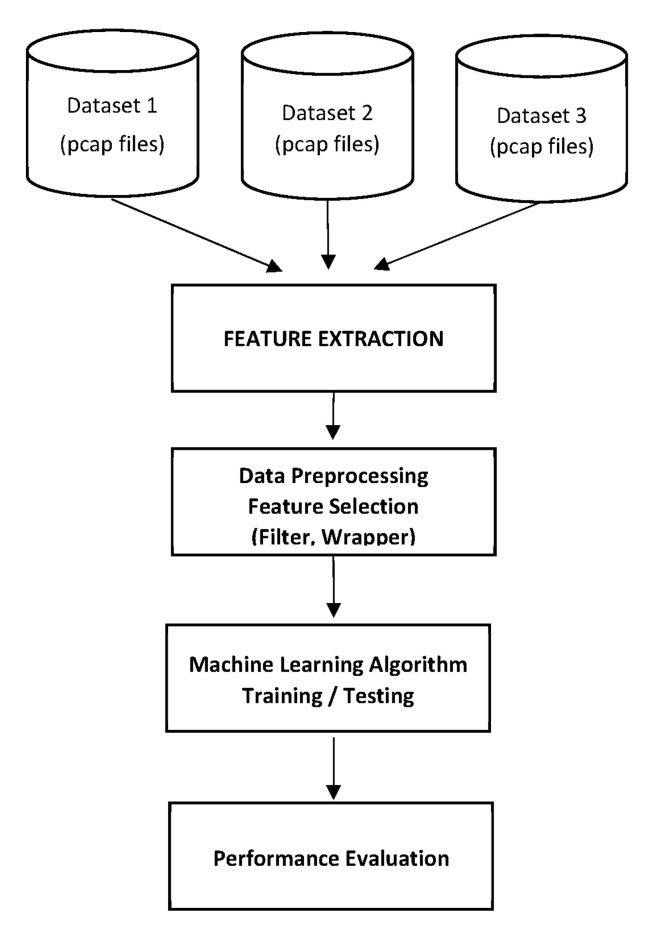
System architecture model.

**Figure 2 sensors-22-03646-f002:**
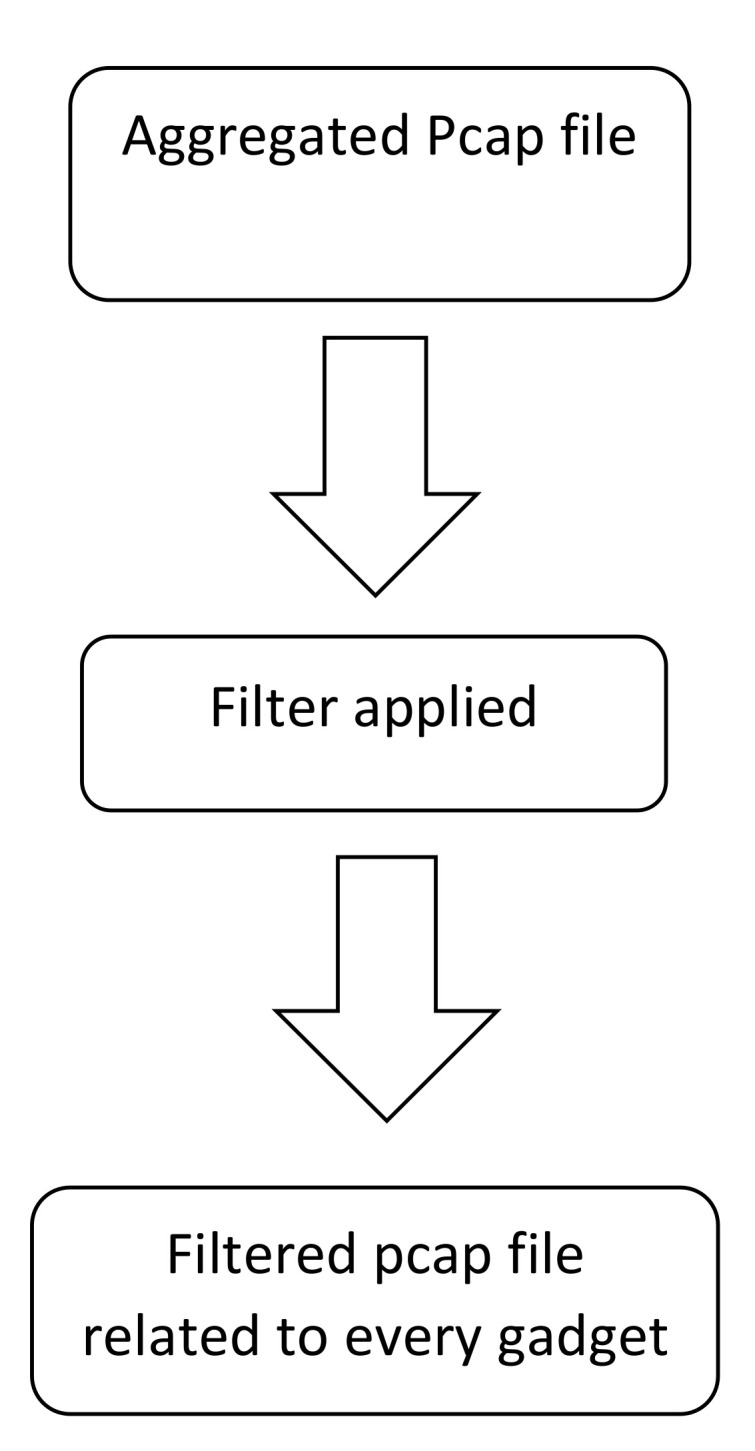
PCAP filtering missing value strategy.

**Figure 3 sensors-22-03646-f003:**
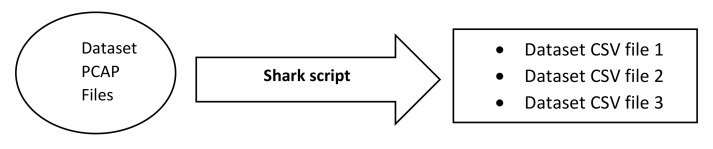
Feature extraction.

**Figure 4 sensors-22-03646-f004:**
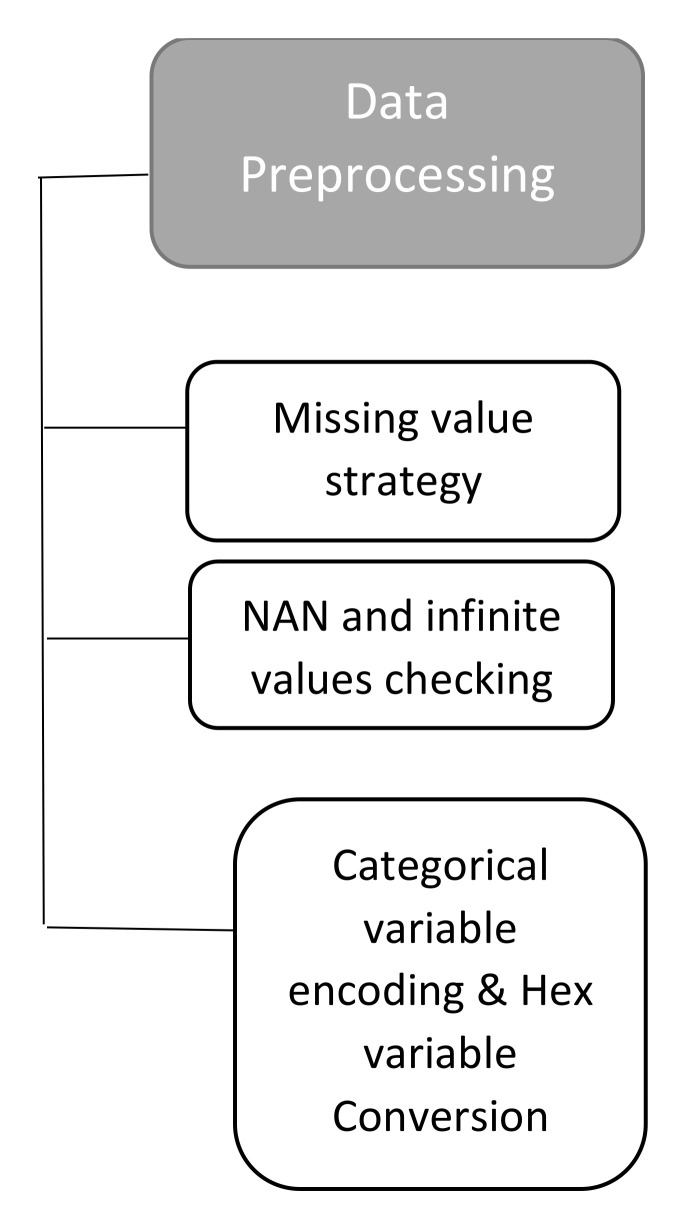
Data preprocessing.

**Figure 5 sensors-22-03646-f005:**
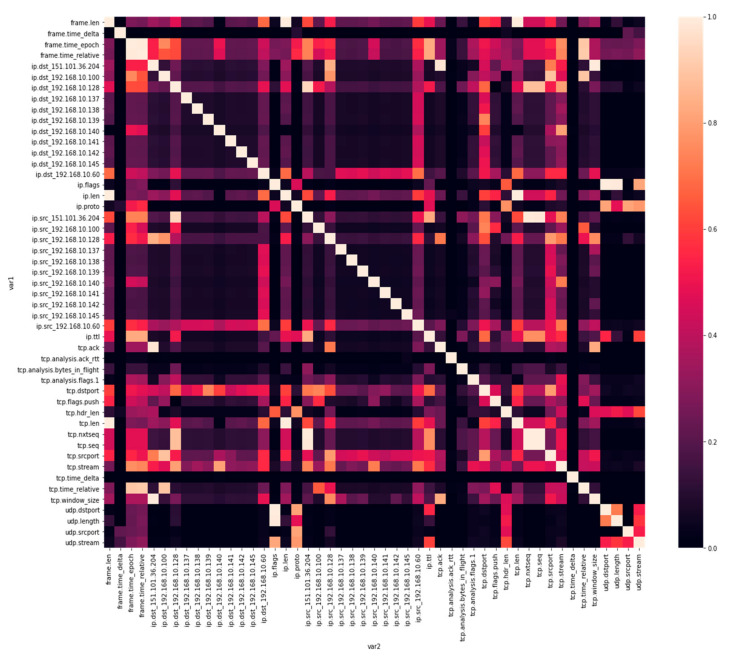
Heatmap showing correlation score.

**Figure 6 sensors-22-03646-f006:**
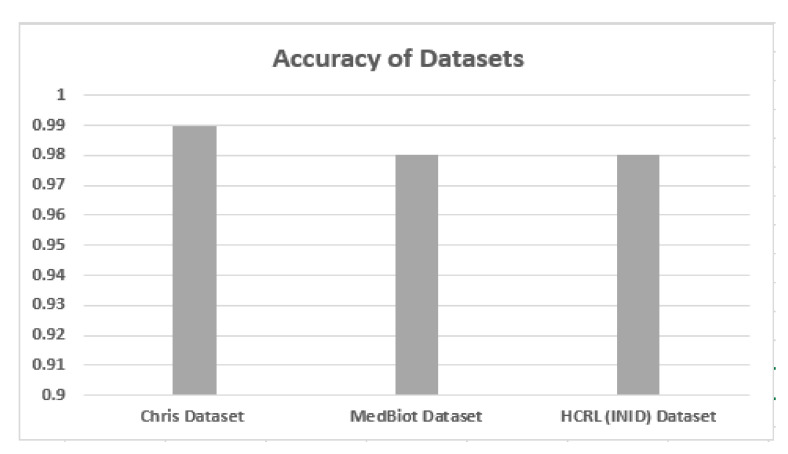
Accuracy of different datasets.

**Figure 7 sensors-22-03646-f007:**
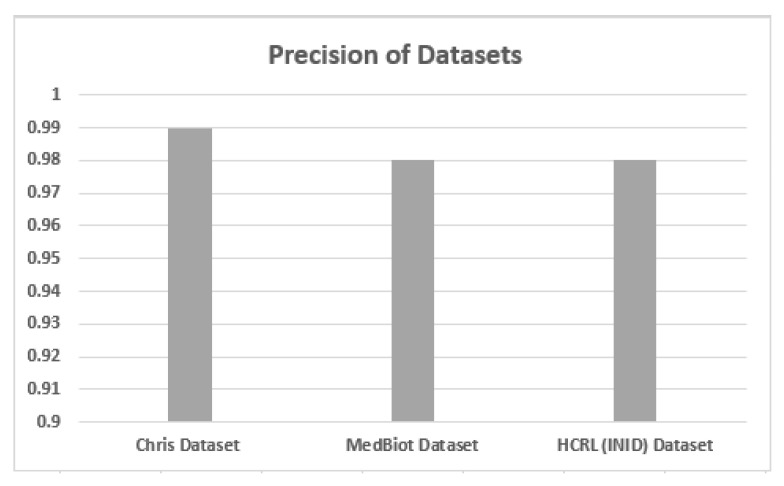
Precision of different datasets.

**Figure 8 sensors-22-03646-f008:**
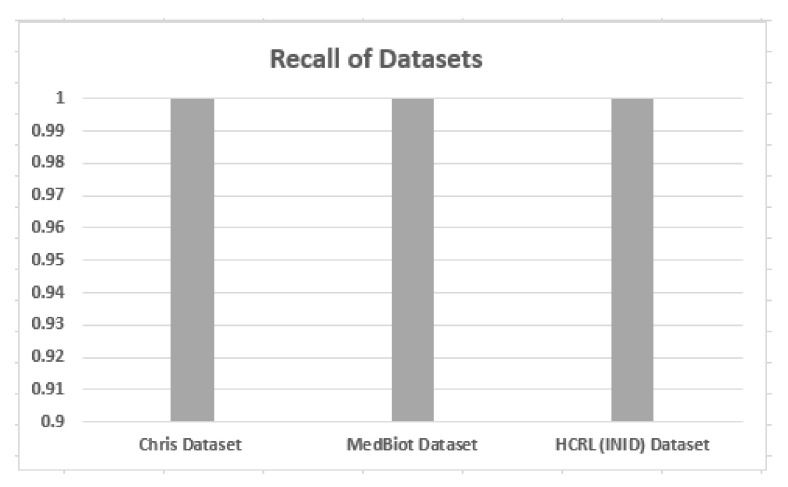
Recall of different datasets.

**Figure 9 sensors-22-03646-f009:**
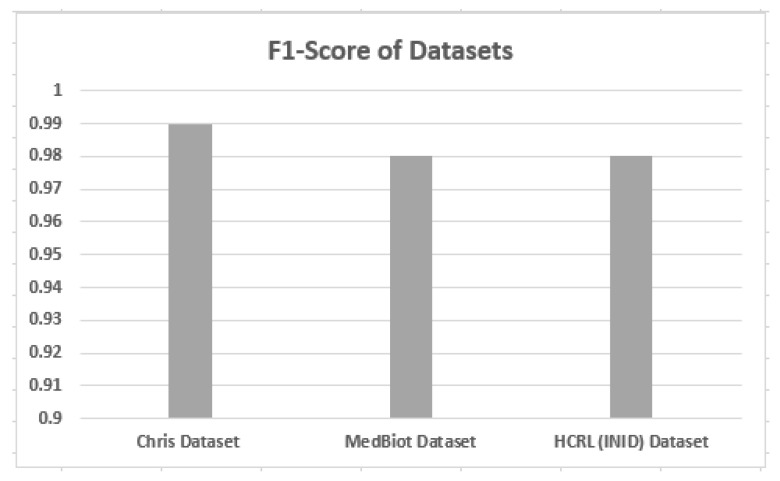
F1-score of different datasets.

**Figure 10 sensors-22-03646-f010:**
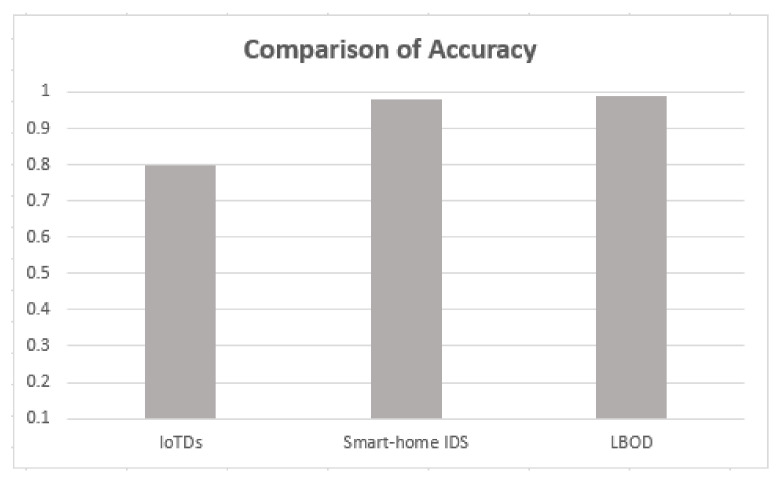
Comparison of accuracy.

**Figure 11 sensors-22-03646-f011:**
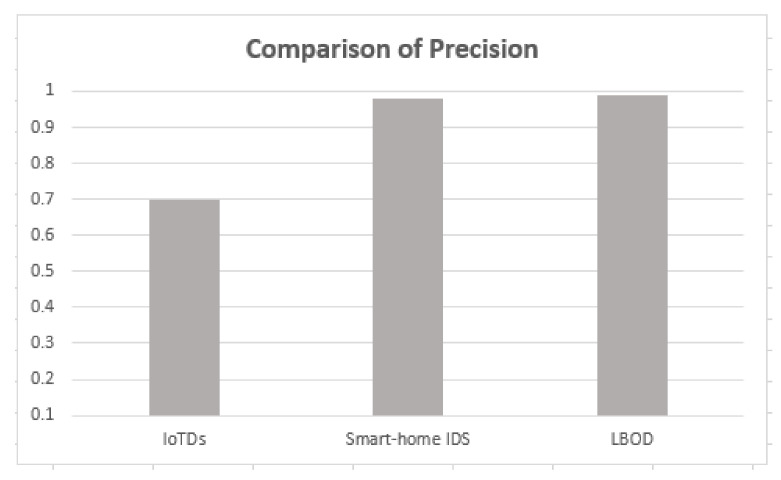
Comparison of precision.

**Figure 12 sensors-22-03646-f012:**
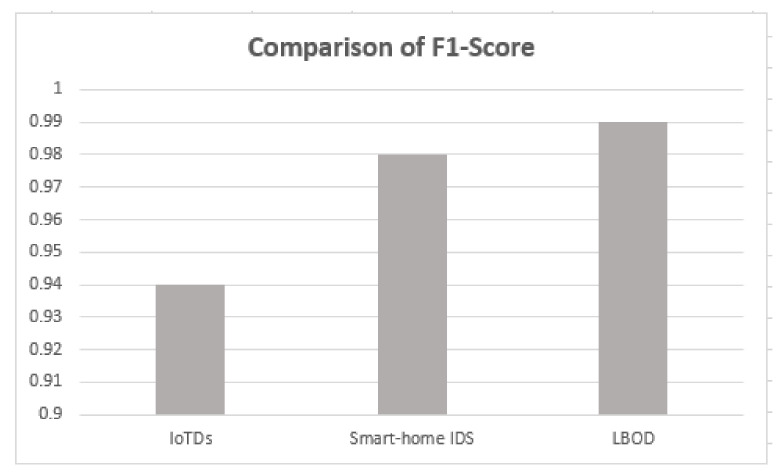
Comparison of F1-score.

**Figure 13 sensors-22-03646-f013:**
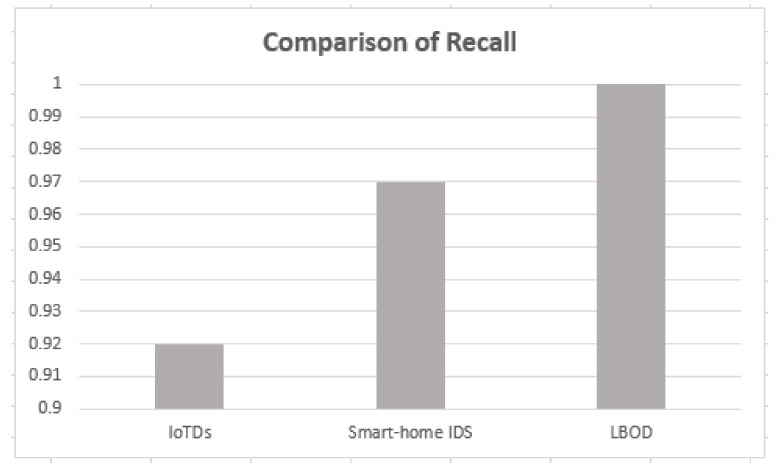
Comparison of recall.

**Table 1 sensors-22-03646-t001:** Extracted features.

S. No.	Feature Name	Description	S. No.	Feature Name	Description
1.	ip.src	Source IP address	24.	tcp.flags.syn	TCP Syn flag
2.	ip.dst	Destination IP address	25.	tcp.flags.ack	TCP ACK flag in packet
3.	frame.len	Length of frame in bytes	26.	tcp.flags.push	TCP PUSH flag in packet
4.	ip.proto	IP protocol number	27.	tcp.flags.reset	TCP RESET flag in packet
5.	tcp.srcport	TCP source port	28.	tcp.flags.fin	TCP fin flag in packet
6.	tcp.dstport	TCP destination port	29.	ip.flags	IP header flags, such as fragmentation
7.	udp.srcport	UDP source port	30.	ip.frag_offset	IP fragmentation flag
8.	udp.dstport	UDP destination port	31.	ip.ttl	Time to live of IP packet
9.	tcp.seq	TCP sequence numbers	32.	tcp.ack	TCP ACK packet of three-way handshake
10.	frame.time_epoch	Packet timestamp	33.	tcp.window_size	Windows size for TCP communication
11.	tcp.stream	TCP streams between nodes	34.	tcp.nxtseq	Next expected sequence number
12.	frame.time_relative	Time since the first packet in frame received	35.	tcp.analysis.flags	Flags for analysis TCP sequence number and Acknowledgment
13.	ip.len	Total length of packet/size of IP frame	36.	udp.stream	Statistics of UDP streams
14.	tcp.len	Length of TCP payload	37.	udp.length.bad	UDP bad length value message
15.	udp.length	Length of UDP payload	38.	udp.length.bad_zero	UPD length is zero
16.	frame.time_delta	Difference time between frames	39.	frame.packet_flags_fcs_length	FCS (frame check sequence) length
17.	ip.hdr_len	Length of IP header	40.	ip.fragment.error	Defragmentation error
18.	tcp.hdr_len	Size of TCP header in 32 bits	41.	tcp.analysis.keep_alive	TCP keep-alive segment
19.	tcp.analysis.bytes_in_flight	Bytes in flight for each packet	42.	tcp.analysis.window_full	TCP windows full specified by user
20.	tcp.time_relative	Time since first frame in TCP session	43.	tcp.analysis.window_update	TCP window update
21.	tcp.time_delta	Elapsed time between current and prior packet	44.	tcp.analysis.zero_window	TCP zero window segment
22.	tcp.analysis.ack_rtt	TCP ack and RTT (round time trip) for packet	45.	tcp.analysis.zero_window_probe	TCP zero window probe
23.	tcp.flags	TCP flags	46.	frame.cap_len	Length of the captured frame

**Table 2 sensors-22-03646-t002:** Features finalized after selection process.

1	frame.len	8	tcp.hdr_len
2	ip.proto	9	tcp.time_relative
3	udp.srcport	10	ip.flags
4	udp.dstport	11	tcp.ack
5	tcp.stream	12	tcp.nxtseq
6	frame.time	13	ip.frag_offset
7	ip.len		

**Table 3 sensors-22-03646-t003:** Dataset description.

Dataset	IoT Devices	Source
MedBiot [27]	IoT device	TPLink smart switch
Chris Dataset [28]	IoT device	Camera
HCRL (INID) [29]	IoT device 1	SKT NUGU (NU 100) Speaker
	IoT device 2	EZVIZ Wi-Fi Camera

**Table 4 sensors-22-03646-t004:** Effect of feature selection.

Data Type	Chris Dataset			MedBiot Dataset			HCRL (INID) Dataset		
	F1-Score	Accuracy	Recall	F1-Score	Accuracy	Recall	F1-Score	Accuracy	Recall
Normal	88%	87%	94%	81%	88%	92%	83%	87%	91%
FS Applied	99%	99%	100%	98%	98%	100%	98%	98%	100%

**Table 5 sensors-22-03646-t005:** Comparison with current research works.

Research	F1-Score	Feature Selection	Multiple Dataset	One-Class Classifier
IOTDS [17]	94%	No	No	Yes
Smart home IDS [18]	98%	No	No	No
Proposed Solution (LBOD)	99%	Yes	Yes	Yes

**Table 6 sensors-22-03646-t006:** Comparison of training and prediction time. FS: feature selection.

Datasets	Chris Dataset		MedBiot Dataset		HCRL (INID) Dataset	
Time (Seconds)	Training	Prediction	Training	Prediction	Training	Prediction
Before FS	5.3457	0.5636	7.5342	2.2327	4.3283	0.7124
After FS	2.2340	0.1298	3.2345	0.4335	2.2134	0.3190

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
