# Peer review of "Lightweight Internet of Things Botnet Detection Using One-Class Classification"

_sensors, 2022, doi:10.3390/s22103646_

Round 1
Reviewer 1 Report
The authors proposed a one-class KNN based botnet detection system that was evaluated on 3 different IoT datasets showing promising results. The paper is well written and the scientifically sound but must be improved before publication.
First major concern is lack of detail or description of the 47 features that are being used for the training of the system. It is important to discuss this for reproduciblity. The features could be outlined in a table and then discussed, and the those that have been subsequently dropped (or selected) during feature selection should be outlined.
Figure 5 must be expanded to make it readable.
In section V.A (Dataset description), more detail should be given about the 3 datasets used. Table 1 does not tell us much about the datasets. At least an informative summary of the 3 datasets should be provided.
The authors also need to specify how many instances of the benign and malicious packets were used in their experiments for each dataset.
The training and testing strategy should specify what percentage of the datasets were used for training and testing respectively.
The authors should proof-read the manuscript to correct minor formatting, grammar and spelling errors. For example:
- Paragraphs 1 and 2 in the introduction should be merged.
- Page 8 need to be re-formatted and made consistent.
- Page 5: correct the statement 'This script was able of producing 47 unique features ....'
- Page 5: correct the statement 'after feature extraction should be gone through the data processing phase as shown in figure 4'
Author Response
Please see the attachment
1. Response to the reviewers

Reviewer 2 Report
The authors need to improve the organization of presentation and technical soundness.
Moreover, there are many flaws that should be addressed.
- All figures should be clear and redrawn with a similar font style.
- In Figure 1, the feature selection process shall be included in the data pre-processing step.
- In Figure 3, the flow or procedure of the feature extraction process should be revised.
- There is no clear why the Hex value conversion needs to have.
- In the sub-section “E. Feature Selection”, the authors mentioned 47 features are extracted in each dataset. Which features did you extract? Are these features the same with different datasets?
- The feature selection processes are also outlandish.
- The accuracy measurements are also springing up in the figures in section V.
- In Table 3, the expression “IDS [18]” is needed to revise.
- The authors state that your solution was able to detect unknown malicious traffic, but it is not possible to guarantee for that with your solution.
- The references provided are applicable, but these are not sufficient. There are too many state-of-art machine learning-based attack detection systems that existed. You will need to add some more related works to highlight your method is better or can fix the gaps of these works.
- Too many typos are needed to fix.
Author Response
Please see the attachment:
1) Response to reviewer

Round 2
Reviewer 1 Report
The paper has been significantly improved, taking into consideration the issues raised in the first round of review. The authors are advised to carry out additional proof-reading the correct the few grammatical errors that are still present within the manuscript.
Author Response
Please see the attached document for author responses.

Reviewer 2 Report
Thanks for your revision. But, some issues are still needed to address.
1) Are the following sentences correct?
- "However, dimensionality reduction process may result in losing of some important features."
- "Filter Method is a method for feature selection that does not use the ML model while choosing features."
2) In Univariate Filter, why do you only import benign file?
3) The authors need to check the following sentences.
• After implementing both the Univariate and Multivariate Filter feature selection strategy, just 22 features were left of initial 46.
• Among two associated features, one was deleted.
• Using the above strategy 21 features were deleted.
4) Why do you use the threshold value as 0.95 for filtering features? As well, why does the gain-value 0.844 become the threshold value in the wrapper-based feature selection process?
5) In 3.6.2, The forward and backward phases are conventional in wrapper-based approaches. They aren't the types of Wrapper methods.
6) In section 4, you mentioned as:
"The One-Class KNN algorithm was chosen as it generates a classification model that can be easily explained, allowing a better interpretation of the classification result. Due to the aforementioned fact, we utilized One-class KNN [36] in our model."
- You may choose lightweight ML algorithms for the lightweight purpose because you expressed that your model is for a lightweight solution.
7) How many percentages of benign records are included in 80 thousand instances?
8) In Table 4, you should add the results with a) the features after filter approach, b) the features after filter & wrapper approaches, because they are separately done by each other.
In addition, it would be better if you compare with other factors (accuracy, recall, ....) rather than only compare with F1-score.
9) The title of sub-section (5.6) should be revised.
10) Double-check typos.
Author Response

(The authors gave the same response as above.)
